# Targeting the DEAD-Box RNA Helicase eIF4A with Rocaglates—A Pan-Antiviral Strategy for Minimizing the Impact of Future RNA Virus Pandemics

**DOI:** 10.3390/microorganisms9030540

**Published:** 2021-03-05

**Authors:** Gaspar Taroncher-Oldenburg, Christin Müller, Wiebke Obermann, John Ziebuhr, Roland K. Hartmann, Arnold Grünweller

**Affiliations:** 1Gaspar Taroncher Consulting, Philadelphia, PA 19119, USA; 2Institute of Medical Virology, Justus Liebig University Giessen, Schubertstrasse 81, 35392 Giessen, Germany; christin.mueller@viro.med.uni-giessen.de (C.M.); john.ziebuhr@viro.med.uni-giessen.de (J.Z.); 3Partner Site Giessen-Marburg-Langen, German Center for Infection Research (DZIF), Hans-MeerweinStrasse 2, 35034 Marburg, Germany; 4Institute of Pharmaceutical Chemistry, Philipps University Marburg, Marbacher Weg 6, 35032 Marburg, Germany; wiebkeobermann@staff.uni-marburg.de (W.O.); roland.hartmann@staff.uni-marburg.de (R.K.H.)

**Keywords:** pan-antiviral, rocaglates, eIF4A, silvestrol, CR-31-B, Zotatifin, translation initiation, coronavirus, COVID-19

## Abstract

The increase in pandemics caused by RNA viruses of zoonotic origin highlights the urgent need for broad-spectrum antivirals against novel and re-emerging RNA viruses. Broad-spectrum antivirals could be deployed as first-line interventions during an outbreak while virus-specific drugs and vaccines are developed and rolled out. Viruses depend on the host’s protein synthesis machinery for replication. Several natural compounds that target the cellular DEAD-box RNA helicase eIF4A, a key component of the eukaryotic translation initiation complex eIF4F, have emerged as potential broad-spectrum antivirals. Rocaglates, a group of flavaglines of plant origin that clamp mRNAs with highly structured 5′ untranslated regions (5′UTRs) onto the surface of eIF4A through specific stacking interactions, exhibit the largest selectivity and potential therapeutic indices among all known eIF4A inhibitors. Their unique mechanism of action limits the inhibitory effect of rocaglates to the translation of eIF4A-dependent viral mRNAs and a minor fraction of host mRNAs exhibiting stable RNA secondary structures and/or polypurine sequence stretches in their 5′UTRs, resulting in minimal potential toxic side effects. Maintaining a favorable safety profile while inducing efficient inhibition of a broad spectrum of RNA viruses makes rocaglates into primary candidates for further development as pan-antiviral therapeutics.

## 1. Introduction

The frequency of infectious disease outbreaks caused by RNA viruses of zoonotic origin in human populations has been rising at an alarming rate in recent years. Prominent examples since the beginning of this millennium are severe acute respiratory syndrome coronavirus (SARS-CoV) (2002–2004), influenza A virus subtype H1N1 (2009–2010), and Middle East respiratory syndrome coronavirus (MERS-CoV) (2012-present). The causes for this intensification are many, including anthropogenic modifications of the environment, encroachment of humans into forested and other natural areas, and overall increased human contact with wildlife [1].

The ongoing SARS-CoV-2 pandemic is the latest example of the widespread and damaging effects such outbreaks can have on a global scale. In just the past three decades, such events have resulted in millions of excess deaths and in economic losses totaling billions of USD [2,3,4]. Preventing new outbreaks from occurring through continuous and coordinated surveillance remains a top public health priority, but developing strategies to mitigate the effects of an outbreak once it is detected, and while a vaccine is developed, has taken on an added sense of urgency [5].

In parallel to the increase in the incidence of infectious diseases, there has been a gradual shift toward more outbreaks being of viral origin rather than caused by other infectious organisms such as bacteria, protozoa, parasites, or fungi [6]. Historically, viruses constituted about one-tenth of all novel human pathogens of zoonotic origin but since 1980 that proportion has increased to about two-thirds [7]. Viruses replicate with extremely high mutation rates compared to other microorganisms, allowing them to rapidly adapt to new hosts and evolve resistance to vaccines and antiviral drugs [8]. Moreover, among viruses, RNA viruses exhibit the highest mutation rates compared to DNA viruses, which gives them an evolutionary advantage that results in RNA viruses accounting for over 80% of the zoonotic viral burden and a disproportionate contribution to the total burden of zoonotic human pathogenesis [9]. Indeed, all major epidemic and pandemic outbreaks since 2000 have been caused by RNA viruses (Table 1). This dominance is most likely not just a result of their high genetic plasticity but also of their high human-to-human transmissibility, especially in cases where RNA viruses can be transmitted by aerosols [10].

Given the disproportionate contribution of RNA viruses to human infectious disease of zoonotic origin and their dominant role in causing major epidemics and pandemics over the past 20 years, developing antiviral therapeutics targeting RNA viruses should be a priority in the quest to curb the spread of future emerging and re-emerging RNA virus outbreaks. Accurate predictions of which RNA viruses will cause future outbreaks is however impossible due to the multiplicity of potential animal hosts and the genetically highly heterogeneous makeup of the virus strains present in these hosts [11]. Ideally therefore, antivirals developed for use against future outbreaks of zoonotic RNA viruses would have to be effective against positive (+)- and negative (−)-stranded RNA viruses across as many virus families as possible.

Antivirals can target either the virus or the host. Virus-specific or direct-targeting antiviral strategies, which include neutralizing antibodies targeting surface antigens of the virus itself, compounds targeting virus–receptor interactions, fusion/budding inhibitors, and viral protease inhibitors [12], are by definition directed against known viral strains, and their development can only be promoted to a preliminary stage before an outbreak happens. Nonspecific virus-targeting antivirals, which consist mostly of nucleoside analogs that inhibit the viral RNA polymerase machinery [12], have found broader application because they can be developed a priori, but their efficacies and safety profiles have been inconsistent and not directly transferable among viral strains. The biggest drawback of direct-targeting antivirals however has been the high selective pressure they exert on the RNA viruses themselves. This results in enhanced mutation rates and translates into unpredictable “moving goalposts” for drug development and, in more extreme cases, gives rise to escape mutants that can render antivirals altogether ineffective over time [13,14,15].

By contrast, host-targeting antiviral strategies, including host receptor inhibitors, endosomal pathway inhibitors, host protease inhibitors, modulators of lipid metabolism, modulators of innate immune response or assorted nuclear signaling pathway modulators [16], minimize the risk of viral mutations by eliminating any direct selective pressure on the viruses. A potential drawback of host-targeting antivirals, however, resides in the risk of substantial side effects caused by interfering with the normal function of the targeted host factors and pathways [11]. Consequently, host-targeted antivirals tend to have narrow therapeutic windows and are complex to manage from a clinical perspective [15].

One host mechanism essential to viral proliferation is translation. RNA viruses do not encode their own translational machinery, rendering them dependent on host protein synthesis. The eukaryotic translational machinery has long been targeted in the context of cancer because aberrant mRNA translation and high expression levels of oncogenes are two hallmarks of many neoplasias [17]. Targeting protein synthesis to inhibit viral proliferation has only been proposed more recently as an attractive therapeutic option to treat viral or bacterial infections [18,19]. The ongoing SARS-CoV-2 pandemic has further accelerated the clinical development of such antivirals through repurposing compounds already in development to inhibit host translation factors in the context of cancer [20,21,22,23]. Importantly, the potential broad-spectrum application of host-specific translational inhibitors is a crucial argument for their development. Such pan-antivirals could be deployed as first-line drugs in the event of an epidemic or pandemic outbreak caused by a novel virus for which there are no direct-targeting antivirals available.

Recent advances in the development of compounds to target the cellular, cap-dependent DEAD-box RNA helicase eIF4A, an essential factor in viral protein synthesis, have underscored the potential of targeting this translational host factor as an antiviral strategy [24]. A large number of RNA viruses depend on eIF4A to translate their mRNAs because the complex structure of their 5′ untranslated regions (5′UTRs) requires the helicase activity of eIF4A to form the 43S-preinitiation complex (43S-PIC) during translation initiation [25]. In vitro, ex vivo, and in vivo inhibition of eIF4A with small natural compounds has been shown to prevent replication of RNA viruses, including corona-, picorna-, flavi-, filo-, hepe-, toga-, arena-, nairo-, and bunyaviruses [26,27].

A very promising class of eIF4A inhibitors are rocaglates, a group of flavaglines that target eIF4A with high specificity [28,29]. The resulting high potencies and optimal selectivity indices compared to all other known eIF4A inhibitors make rocaglates ideal candidates for further preclinical and clinical development as pan-antivirals.

In this review, we discuss the essential role of eIF4A in RNA virus translation, the antiviral properties of all known eIF4A inhibitors, recent advances in our understanding of the rocaglate-based eIF4A inhibition mechanism, and the broad spectrum of rocaglate-mediated eIF4A antiviral activities, and we lay out a roadmap for advancing rocaglates through preclinical and clinical development. Our ability to reduce the widespread and damaging effects of future epidemics and pandemics will greatly benefit from having access to pan-antiviral drugs that can help manage the initial phases of an outbreak.

## 2. The Role of the DEAD-Box RNA Helicase eIF4A in RNA Virus Translation

All known viruses usurp the translational machinery of their host cells to synthesize large amounts of viral proteins in a short period of time. To achieve this, viral mRNAs compete with the host’s own mRNAs for access to the necessary factors of the eukaryotic translation initiation apparatus, a complex machinery comprising at least twelve different components [30]. Indeed, all viruses, but RNA viruses in particular, have developed several sophisticated strategies to direct the host’s translation machinery to preferentially synthesize viral proteins over the host’s own proteins [31,32,33]. The two translation mechanisms mainly used by RNA viruses are cap-dependent translation and internal ribosome entry site (IRES)-dependent translation.

Cap-dependent translation follows a precise sequence of events orchestrated by different eukaryotic initiation factors (eIFs) [34]. First, the mRNA is recruited to the heterotrimeric eIF4F complex [35]. This complex consists of the cap-binding protein eIF4E, eIF4A, and the scaffold protein eIF4G that binds to the poly (A)-binding protein (PABP) and eIF3, another initiation factor [36,37]. Binding of eIF4F to the cap structure leads to cyclization of the mRNA [38]. Next, and in order for the 40S ribosomal subunit to gain access to the highly structured viral 5′UTRs, eIF4A springs into action [39]. The helicase unwinds RNA secondary structures in the 5′UTR and removes adherent proteins to generate an unstructured region that allows stable binding of the 43S-PIC [40,41]. This complex scans the 5′UTR to identify the AUG start codon where formation of the elongation-competent 80S complex takes place followed by protein synthesis [38,41,42,43,44] (Figure 1).

In IRES-dependent translation, the 40S ribosomal subunit is recruited directly to the mRNA start codon by binding to the secondary structure of an IRES [45]. Most viral IRES-dependent translation mechanisms have been categorized into one of four classes based on the RNA structures involved and the initiation factors they recruit [46]. In classes I to III, the IRES is located in the 5′UTR and translation initiation depends on eIFs, while class IV IRES are located in intergenic regions and do not recruit any eIFs. Class I and II IRES contain simple short and long hairpin structures while class III IRES contain more complex, knotted secondary structures. Class I and II IRES-dependent translation requires a combination of eIF4G, eIF3, eIF2, and eIF4A, while class III does not use eIF4A to assemble the 40S initiation complex [46,47,48]. Finally, ribosome recruitment occurs upstream of the start codon in class I IRES-dependent translation and requires 5′ to 3′ scanning to reach the start codon, while the translation complex binds directly to the start codon in class II IRES-dependent translation [46].

The translation initiation factor eIF4A is a DEAD-box RNA helicase, a group of ATP-dependent eukaryotic RNA helicases named after the conserved amino acid sequence Asp-Glu-Ala-Asp (D-E-A-D) [49]. There are three paralogs of eIF4A in mammals: eIF4A1 (DDX2A), eIF4A2 (DDX2B), and eIF4A3 (DDX48). While eIF4A1 and eIF4A2 have a sequence identity of 90–95%, eIF4A3 shares only ~60% sequence identity with eIF4A1 [50]. Functionally, eIF4A3 differs from eIF4A1 and eIF4A2. eIF4A3 is involved in the assembly of the Exon Junction Complex, which coordinates splicing of pre-mRNAs [50]. By contrast, eIF4A1 and eIF4A2 exhibit equivalent biochemical activities but differ significantly in biological function and expression levels in vivo. eIF4A1 is present in almost all tissues during active cell growth, whereas eIF4A2 is mainly produced in organs with low proliferation rates [51]. In addition, when eIF4A1 is suppressed, eIF4A2 levels rise, but the loss of eIF4A1 is not fully compensated. Consequently, eIF4A1 but not eIF4A2 is essential for cell survival.

All DEAD-box RNA helicases contain two RecA-like domains that are connected by a flexible linker [52]. In the absence of an RNA substrate or ATP, the two domains adopt an inactive open conformation [53,54]. Binding to RNA and ATP leads to closure of the gap between the two domains (Figure 2). In this closed, active conformation, the conserved motifs of the two domains form an interface exhibiting ATPase and helicase activities. Following ATP hydrolysis, the gap between the two domains re-opens to allow the release of the unwound RNA.

While eIF4A is the main helicase responsible for unwinding RNA secondary structures in 5′UTRs, several other RNA helicases play essential roles during translation [55]. Among them, the RNA helicase DEAD-box polypeptide 3 (DDX3) was reported to facilitate translation of complex secondary RNA structures in general as well as of secondary structures specifically associated with the 7-methylguanylate structure (m^7^GTP) of the RNA cap [46]. DDX3 also plays a role equivalent to that of eIF4A in class I and II IRES-dependent translation [46], and its essential role in viral RNA translation has been documented for a number of viruses, including the RNA viruses Japanese encephalitis virus, Dengue virus (DENV), and West Nile virus [56,57,58,59]. Several studies have further shown the broad-spectrum antiviral potential of targeting DDX3 [60,61,62].

## 3. eIF4A Inhibitors

A number of natural products that inhibit protein synthesis in eukaryotic cells, and eIF4A in particular, have been described, and their number continues to grow [63,64] (Figure 3). Originally, eIF4A inhibitors were identified as potential antitumor therapeutics [65,66]. In addition to its ATP-binding pocket, eIF4A has a nucleic acid-binding region where RNA substrates bind via their phosphate backbone in a sequence-independent manner, providing several possible interaction surfaces for inhibitors to bind to the eIF4A–RNA complex [67]. High-throughput screens for eukaryotic translation inhibitors resulted in the identification of three natural substances—hippuristanol, pateamine A (PatA), and silvestrol—that differ substantially in their chemical structures [68,69,70].

Hippuristanol is a polyhydroxysteroid found in the golden fan coral *Isis hippuris* (Figure 3). Hippuristanol interacts with the C-terminal domain of eIF4A via motifs V and VI (Figure 2), preventing the binding of RNA [71]. Through this allosteric inhibition, eIF4A is fixed in the closed conformation and cannot release the unwound RNA substrate [72]. Hippuristanol is a selective inhibitor of eIF4A due to the high sequence variation of motifs V and VI across DEAD-box helicases [72]. In contrast to RNA binding, the binding of ATP can take place in the presence of the inhibitor because the N-terminal domain of eIF4A is not affected by hippuristanol [71]. The antiviral activity of hippuristanol has been documented against several viruses such as the encephalomyocarditis virus (EMCV) and the norovirus, two positive-stranded RNA viruses, and human T cell leukemia virus type 1 (HTLV-1), a retrovirus [70,73,74].

PatA is a macrolide isolated from the encrusting marine sponge *Mycale hentscheli* (Figure 3) that induces dimerization of eIF4A and RNA [69,75,76]. PatA disrupts interaction with eIF4G and reduces levels of eIF4A present in the eIF4F complex [68,75], which in turn may affect the assembly of the 43S-PIC. In contrast to hippuristanol, PatA only binds free eIF4A, suggesting that the binding site for PatA likely occurs at the interface of the eIF4A N- and C-terminals domains, two domains rendered inaccessible once the eIF4F complex has formed [66]. PatA has been shown to have antiviral activity against influenza A virus, a negative-stranded RNA virus [77].

Silvestrol is a rocaglate isolated from Asian mahogany plants of the genus *Aglaia* [28] (Figure 3). A characteristic feature of silvestrol and its diastereomer episilvestrol is the presence of a 1,4-dioxane moiety linked to their A rings (Figure 3 and Figure 4) [29]. Since the first rocaglate, rocaglamide A (RocA), was isolated and its chemical structure solved, over 200 rocaglates have been identified [78] and isolated, including silvestrol and episilvestrol in 2004 [79,80]. A chemical synthesis route of RocA was published in 1990 that allows the control of the absolute stereochemistry of the molecule class [81]. Since then, the synthesis has been further optimized and expanded to include a large number of modified rocaglates [82]. An example of such a non-naturally occurring rocaglate is CR-31-B. The synthesis of CR-31-B results in a racemic mixture of two enantiomers, of which only the (−)-enantiomer is biologically active [83,84]. The antiviral activity of natural and synthetic rocaglates has been well documented (see Section 5 below).

In addition to the three best-known classes of eIF4A inhibitors, other low-molecular-weight compounds have been shown to inhibit the helicase, although the specificity and selectivity of several of them remains to be established, and their potential antiviral activity has not yet been determined [85]. The list includes allolaurinterol, elatol, elisabatin A, 6-aminocholestanol, sanguinarine, and the prostaglandin 15d-PGJ2 [86,87,88,89,90]. Allolaurinterol and elatol are found in red algae and silvestrol and other rocaglates in plants, all within the supergroup *Archaeplastida*. By contrast, hippuristanol and elisabatin A have been isolated from cnidaria (corals) and pateamine A from porifera (sponges) species, both in the supergroup *Opisthokonta*. Finally, 15d-PGJ2 is an example of an eIF4A inhibitor that is produced in mammalian (human) cells [88]. Considering that eIF4A, the prototype of DEAD-box RNA helicases [49], is an evolutionarily ancient and highly conserved enzyme, the widespread occurrence of chemically diverse eIF4A inhibitors across eukaryal supergroups suggests that the potential to interfere with eIF4A activity evolved independently as an advantageous antagonistic principle in eukaryal evolution (Figure 3).

## 4. Mechanism of Action of Rocaglates

Rocaglates are a group of flavaglines that contain a characteristic cyclopenta[*b*]benzofuran structure (Figure 4) [28,29].

Initial binding studies of eIF4A with several rocaglates—and the more recent elucidation of the crystal structure of a truncated version of human eIF4A (PDB: 5ZC9) in complex with RocA, polypurine RNA, and AMP-PNP, a non-hydrolyzable ATP analogue—have revealed that rocaglates reversibly clamp the RNA–helicase complex (Figure 5) [65,91,92]. Specifically, RocA forms stable π–π interactions with the phenylalanine residue at position 163 (Phe163) and two consecutive purine bases (5′-AG) in the eIF4A-(AG)_5_RNA complex. Additional hydrogen bonds with Gln195 and Asp198 of eIF4A, as well as with the N7 nitrogen of the guanine base of the RNA substrate, stabilize the RocA–eIF4A–RNA complex [91]. The co-crystal structure has provided the foundation for the structure-based development of new potential eIF4A inhibitors. Comparative in silico docking analysis of silvestrol and CR-31-B (-), two rocaglates with similar antiviral activities, indicates subtle differences in the binding mode between dioxane-containing silvestrol and rocaglates lacking the dioxane moiety [83]. In silico docking results suggest an expanded interface involving contacts of the dioxane moiety of silvestrol to nearby arginine residues of eIF4A (Figure 5). This differential interaction with the eIF4A–RNA complex might explain why “larger” dioxane-containing rocaglates can clamp eIF4A–RNA complexes containing RNA substrates with short hairpin structures, while rocaglates without a dioxane moiety strictly require RNA substrates with unstructured polypurine sequences for complex formation [83]. Ligand-based optimization studies should help further to optimize the clamping characteristics of novel rocaglates [23,93].

DDX3, the second DEAD-box RNA helicase targeted by rocaglates [94], interacts with multiple viral components and affects several processes, including translation [95]. As a result, dual targeting of DDX3 and eIF4A might enhance the broad-spectrum antiviral effect of rocaglates. Mechanistically, there is no evidence for a π–π-stacking interaction with an aromatic residue of DDX3 that may resemble the ring C stack with Phe163 of eIF4A. However, Gln360 of DDX3, analogous to Gln195 of eIF4A, forms an essential hydrogen bond with the rocaglate [94].

At the cellular level, ribosome profiling experiments have revealed the effects of eIF4A inhibition by rocaglates on the cell’s transcriptional program. Several studies showed that the global cellular effects of eIF4A inhibition are limited to about 300 cellular mRNAs [96,97]. Among the affected mRNAs, proto-oncogenic mRNAs with relatively long and structured 5′UTRs prevailed, which explains the strong antitumor effects of rocaglates [28,79]. This mRNA selectivity also might explain the low toxicity of rocaglates in primary cells and animals [98]. The cytotoxic effects of rocaglates observed in primary cells might be correlated with cellular proliferative activity [99].

## 5. eIF4A as an Antiviral Rocaglate Target

RNA viruses have evolved different ways to capitalize on their host’s translation machinery, making it a challenge to develop a universal strategy for inhibiting viral replication via translation inhibition. However, because of its central role in the translation mechanism, and especially in viral protein synthesis, eIF4A provides an excellent broad-spectrum target in the context of host-targeting antiviral strategies [27]. Next, we will summarize the antiviral effects of rocaglate eIF4A inhibitors in different viral families (see also Figure 6).

### 5.1. Negative (−)-Stranded and Ambisense RNA Viruses

*Arenaviridae*, *Nairoviridae*, *Filoviridae* are negative-stranded (including ambisense) RNA viruses and the causative agents of diseases such as measles, rabies, influenza, and viral hemorrhagic fevers (VHFs). The latter are febrile illnesses caused by members of various (−)-stranded RNA virus families, including arenaviruses (e.g., Lassa virus (LASV)), nairoviruses (e.g., Crimean–Congo hemorrhagic fever virus (CCHFV)), and filoviruses (e.g., Ebola virus (EBOV)) [103]. With fatality rates of up to ~35% to 80% in humans, EBOV is the most prominent member of this group [104]. EBOV comprises seven transcriptional units that possess long and structured UTRs in their 5′ regions, making eIF4A helicase inhibition an obvious choice to combat this virus [105]. Indeed, in EBOV-infected Huh-7 cells and primary macrophages, silvestrol levels of 10–50 nM inhibited viral replication efficiently [18]. Similarly, low nanomolar concentrations of silvestrol and CR-31-B (-) also inhibited proliferation of LASV and CCHFV [83].

### 5.2. Positive (+)-Stranded RNA Viruses

#### 5.2.1. *Coronaviridae*

This large and genetically divergent family of plus-stranded RNA viruses infects a wide range of animals and has significant zoonotic potential, as illustrated by several coronavirus epidemics since the beginning of the century and the current SARS-CoV-2 pandemic [106] (see Table 1). Members of the family *Coronaviridae* are very sensitive to eIF4A inhibition. For example, dose-dependent inhibition by silvestrol and CR-31-B (-) was shown for human coronavirus 229E (HCoV-229E), MERS-CoV, and SARS-CoV-2 replication in vitro, with EC_50_ values in the range of 1–3 nM [26]. In an ex vivo bronchial epithelial cell system, HCoV-229E and SARS-CoV-2 replication was reduced to undetectable levels in the presence of 100 nM CR-31-B (-) [100]. A comprehensive analysis of the SARS-CoV-2 protein interaction map published in early 2020 identified the host translational machinery, and in particular eEF1A and eIF4A, as top targets for drug repurposing aimed at abolishing SARS-CoV-2 replication [107]. Zotatifin, another eIF4A inhibitor of the rocaglate family, was shown to inhibit SARS-CoV-2 replication with an IC_90_ of 37 nM in vitro [107]. Zotatifin entered a phase 1 clinical trial in patients with COVID-19 in November 2020 [22,23,108].

#### 5.2.2. *Togaviridae*

Chikungunya virus (CHIKV), an arthropod-borne virus, is a member of the family *Togaviridae*. The family includes arthropod-borne viruses (arboviruses) that are transmitted by blood-feeding vectors including mosquitos, sandflies, and ticks [109]. The (re)emergence of arboviruses such as CHIKV has been linked to intensive growth of global transportation, arthropod adaptation, and increase of population density due to urbanization. This combination of factors is generally thought to cause an increasing number of arbovirus epidemics [110]. To date, no antivirals are available to treat these emerging arbovirus-related infections effectively [111].

CHIKV RNAs include 5′ capped UTRs with distinct secondary structural elements [112]. Silvestrol treatment of CHIKV-infected 293 T cells was shown to cause delayed viral protein synthesis, a less profound CHIKV-induced host protein shut-off, and suppression of innate immune responses, with an EC_50_ of 1.9 nM [101]. Efficient transmission of CHIKV from infected animals or humans to mosquitoes requires high virus titers [113]. A potent antiviral therapy leading to decreased viral loads could help prevent the sickness and reduce the transmission of CHIKV and potentially other arboviruses in human populations.

#### 5.2.3. *Hepeviridae*

Hepatitis E virus (HEV) is the causative agent of hepatitis E and a common cause of acute viral hepatitis in humans. HEV requires several host eIFs, including eIF4A, eIF4G, and eIF4E for viral mRNA translation [114]. In vitro, silvestrol exhibits a pan-genotypic antiviral effect against HEV [102,115]. In humanized mice, intraperitoneal treatment with 0.3 mg/kg silvestrol inhibited HEV growth with no effect on body weight after 10 days of treatment [102].

#### 5.2.4. *Picornaviridae*

*Picornaviridae* is a family of plus-strand RNA viruses that includes important human pathogens, such as poliovirus (PV) type 1, 2, and 3 and human rhinoviruses (HRV). Picornavirus RNA translation is mediated by class I and II IRES structures and thus depends on eIF4A. In HeLa cells, hippuristanol slowed down PV replication [70], while silvestrol inhibited PV and HRV replication in MRC-5 cells, with EC_50_ values of 20 nM and 100 nM, respectively [83]. The eIF4A-directed antiviral potency of rocaglates was less pronounced in members of the *Picornaviridae* compared to other RNA viruses, such as members of the family *Coronaviridae*. The question of whether or not these differences are related to mechanistic differences between cap-dependent and IRES-dependent eIF4A functions and the relevance of DDX3 with its equivalent role to that of eIF4A in class I and II IRES-dependent translation remains to be addressed in further studies.

#### 5.2.5. *Flaviviridae*

The family *Flaviviridae* includes viruses that employ fundamentally different strategies to ensure viral genome translation. One group of viruses, including members of the genera *Hepacivirus* (e.g., hepatitis C virus (HCV)), *Pestivirus* (e.g., classical swine fever virus (CSFV)), and *Pegivirus*, control viral protein translation strictly through class III IRES elements and are thus resistant to eIF4A inhibitors such as hippuristanol and silvestrol [26,116]. The second group, including arboviruses of the genus *Flavivirus* (e.g., Dengue virus (DENV) and Zika virus (ZIKV)), was widely accepted to employ 5′ cap-dependent translation initiation mechanisms [116]. However, more recent studies suggest that protein synthesis in DENV- and ZIKV-infected cells does not stringently depend on 5´ cap structure-dependent translation, and there is evidence that the 5′UTRs of these viruses have IRES competence [117,118]. Treatment of ZIKV-infected A549 cells and primary human hepatocytes with silvestrol or CR-31-B (-) was reported to inhibit viral replication [119].

Altogether, the broad-spectrum of eIF4A-dependent RNA viruses that are sensitive to rocaglate-mediated inhibition of viral protein translation illustrates the potential of this class of compounds as pan-antiviral treatments in future outbreaks of newly emerging or re-emerging RNA viruses.

## 6. A Pan-Antiviral Translational Roadmap for Rocaglates

The a priori development of pan-antivirals has become increasingly urgent in light of the accelerating pace of novel and re-emerging viral outbreaks [120,121]. Among the host factors known to be required for the replication of a broad-spectrum of viruses, eIF4A stands out as one of the most promising targets. Additionally, rocaglates are increasingly being recognized as some of the most promising broad-spectrum antiviral compounds modulating the activity of eIF4A [24,93]. However, the development of such compounds, from basic and preclinical research to clinical studies and commercialization, involves a number of experimental, monetary, clinical, manufacturing, and regulatory challenges.

### 6.1. Foundational Research

To date, the default rapid response to an outbreak caused by an unknown virus has been to set up large screens of approved antivirals or compounds that have passed phase 1 or 2 (proof-of-concept) clinical trials and have been shown to have favorable safety profiles in humans [122,123,124]. However, no systematic efforts to drive early discovery of novel antivirals and their follow-up development exist.

A framework for the systematic discovery and development of pan-antiviral rocaglates would need to include a large but defined number of in vitro and ex vivo studies showing consistent antiviral effects across viral families and eIF4A-dependent translation mechanisms. Such a framework would provide a solid mechanistic foundation to help predict whether a novel viral strain might be sensitive to rocaglates following an initial sequence-based determination of its putative translation mechanism [125,126]. The framework would ideally also include comprehensive pharmacological profiles of candidate rocaglates, and in particular of their immunomodulatory effects, to maximize treatment safety during a potential emergency rollout in the initial phases of an outbreak [99,127]. A consensus on what the minimum validation criteria of such foundational research need to be would provide the basis for proceeding to preclinical in vivo studies.

### 6.2. Preclinical Research

Once reasonable in vitro and ex vivo evidence of the pan-antiviral activity of a rocaglate exists, critical in vivo studies need to be carried out in preparation for potential clinical development of the compound [128]. The challenge faced here is two-fold: on the one hand, relevant and informative animal models for every representative viral family and/or strain need to be used to demonstrate the pan-antiviral potential of the compound. On the other hand, preclinical studies of drug metabolism and pharmacokinetics need to be carried out to determine the potential viability of a compound as a drug. The latter studies consist of determining the absorption, distribution, metabolism, elimination, and toxicity (ADMET) characteristics of a compound in the context of the animal model used [129]. Initial studies of the pharmacokinetic properties of silvestrol, for example, have shown 100% systemic availability with good distribution to liver, spleen, kidney, but not brain, following intraperitoneal administration. By contrast, oral bioavailability was below 2%, and plasma stability following intravenous administration was about 75% [130]. Extrapolating ADMET values to humans based on accumulated pharmacological knowledge for other drugs is a key component of designing clinical trials.

The need to use a diverse panel of preclinical animal models to test for pan-antiviral activity of a rocaglate means that there will be an equally diverse number of ADMET data that will require careful harmonization to be able to meaningfully extrapolate them to humans. Critical aspects include optimization of the in vivo panel to minimize the number of animal models used and to maximize the similarity of their infection mechanism parameters to humans. If there is for any given virus a choice of model, the model that closest resembles human biology should be used—usually, this would mean to choose larger non-rodent rather than smaller rodent animals. However, this guideline would need to be balanced by how well established and efficient larger infection models are for a given virus, and the ethical, operational, and cost constraints of testing such animals.

### 6.3. Clinical Studies

Once reasonable evidence has been gathered, and the necessary preclinical ADMET data have been collected and evaluated for a range of viral families, the stage is set for carrying out the necessary clinical studies leading to the potential approval of a rocaglate-based antiviral drug. The ongoing SARS-CoV-2 outbreak triggered an unprecedented initial effort to rapidly identify antiviral drugs that could be used to curb infection and/or mortality rates. Although, to shortcut the long time frames required for discovery of novel activities, the immediate focus was on the identification of compounds that had already passed phase 1 clinical studies and for which sufficient safety and potential dosage data in human were available to allow large scale phase 2 and phase 3 clinical trials to take place immediately [122,131]. A similar approach—consisting of keeping an updated database of such safe-in-human compounds with antiviral activities that can be used to rapidly identify drug candidates against a novel virus outbreak—has been proposed as a strategy for the development of antivirals to tackle future outbreaks [132].

Rocaglates offer an opportunity to expand this approach and develop a novel framework for the development of truly pan-antiviral compounds that could be enlisted immediately upon an outbreak of a novel virus to mitigate its initial spread and effects while more permanent solutions such as vaccines are developed. Critical to this framework would be the confirmation that the compound is safe and effective against a broad-spectrum of existing viruses. Once a novel virus appears, such a drug could be immediately deployed and its effectiveness and possible virus-related adverse effects monitored using the existing framework of a phase 4 or so-called post-marketing clinical study [133].

With Zotatifin (eFT226), the first eIF4A inhibitor in the rocaglate family to enter clinical trials as an antiviral, the foundation is set for the further development of rocaglates as potential pan-antivirals [22]. Although at the time of this writing antiviral preclinical data for Zotatifin have not yet been published, the quick progression from initial identification of Zotatifin as an antiviral agent against SARS-CoV-2 to phase 1 clinical studies suggests favorable ADMET results and bodes well for rocaglates overall as a novel class of antivirals.

### 6.4. Other Considerations

Beyond the focus on gathering biological, mechanistic, and clinical evidence of the pan-antiviral potential of a rocaglate, several other critical aspects of drug development need to be considered as well. (i) Manufacturing and scale-up: Rocaglates are complex to manufacture, and as such, the feasibility of manufacturing a lead rocaglate efficiently and in high quantities needs to be determined early in the development process [82]. Key parameters are consistency and scale-up capacity. These parameters are critical when it comes to ease of distribution and the ability to meet demand during an outbreak. Stability and cost are also central to the ability of stockpiling the drug in anticipation of a possible outbreak [134]. (ii) Regulatory: The development of novel antiviral drugs presents several challenges that have been reviewed elsewhere [134]. Of relevance to rocaglate-based antivirals are the complexity of clinical trial design—access to infected populations, ethical issues with placebo treatment in the case of high mortality viral outbreaks—and the challenges associated with determining endpoints—viral load, symptoms, survival.

It is advisable to initiate conversations with industry partners and regulatory authorities early in the process to explore solutions and optimize the overall process. Pharma partners can contribute both invaluable medicinal chemistry knowhow and the infrastructure and resources for clinical trials, while early contact with regulatory bodies can be key to designing and implementing the clinical trials most conducive to eventual approval of the drug.

Finally, before becoming a focal point as potential antiviral drugs, rocaglates were already being extensively researched in the context of cancer [28,98]. In vivo activity of rocaglates has been shown in a number of relevant animal models of cancer, and their ADMET profiles have been well described. In addition to a phase 1 clinical study for SARS-CoV-2, Zotatifin is also in a phase 1–2 clinical study in advanced solid tumors. The body of research accrued in the context of cancer could prove to be very useful in terms of helping accelerate the development of novel rocaglates as antivirals. Given that most infections caused by recent viral outbreaks are acute and respiratory in nature, there is the added advantage of potentially larger therapeutic windows due to short treatment needs and potential local oral administration rather than systemic.

## 7. Outlook and Perspectives

Rocaglates have emerged as strong contestants in the race to develop broad-spectrum antivirals to deploy against future emerging and re-emerging RNA viral outbreaks. Key to their favorable therapeutic profile is a distinct mechanism of action that provides exquisite target specificity and minimal interference with host biology, resulting in potentially large therapeutic windows.

It is worth pointing out here that beyond their antiviral and antitumor activity, rocaglates have also been shown to have eIF4A-dependent antiplasmodial effects in vitro and in vivo against *Plasmodium falciparum* and *P. berghei* [135] and an eIF4A-dependent antifungal effect in vitro against *Candida auris* [136], further highlighting the broad therapeutic potential of this class of eIF4A-targeting compounds against potential pathogens.

With several rocaglates now entering preclinical development and at least one rocaglate already in early clinical development, a translational path for rocaglates as antiviral and eventually pan-antiviral therapeutics is starting to materialize. Many biological, clinical, and practical challenges remain, but a concerted effort to further develop the many therapeutic possibilities of this promising family of compounds should result in an improved understanding of the biology of eIF4A and of the potential benefits of modulating its activity using small molecules.

## Figures and Tables

**Figure 1 microorganisms-09-00540-f001:**
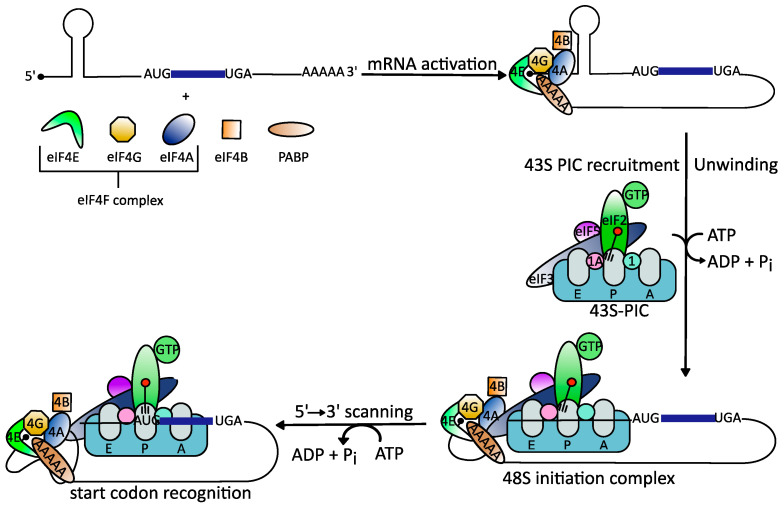
Schematic illustration of the eukaryotic translation initiation mechanism (modified from [38]). Binding of the heterotrimeric eIF4F complex to the 5′ cap structure of mRNAs is followed by unwinding of stable RNA secondary structures by the DEAD-box RNA helicase eIF4A. This enables binding of the 43S preinitiation complex (PIC), which scans down the 5′-untranslated regions (5′UTRs) to identify the start codon AUG.

**Figure 2 microorganisms-09-00540-f002:**
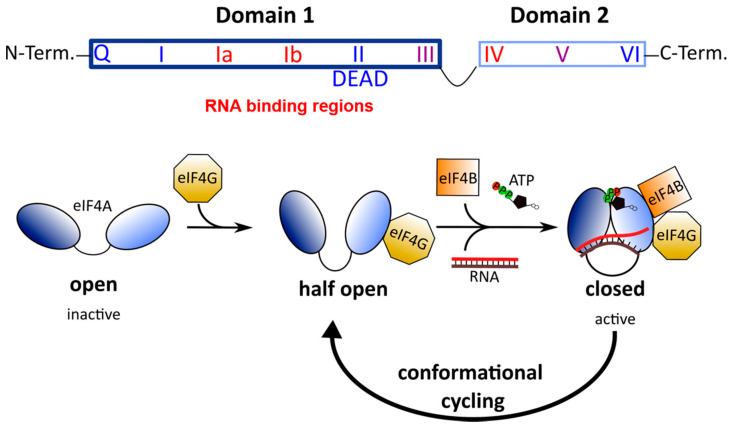
Top: Architecture of DEAD-box RNA helicases. Two RecA-like domains are connected by a flexible linker to form the helicase core. This core consists of 9 conserved motifs that are involved in ATP binding and hydrolysis (blue) and RNA binding (red) [50]. Bottom: Conformational cycling of eIF4A. Binding of eIF4G enables eIF4A to switch from the open to the half-open state. In the presence of eIF4B, ATP and an RNA substrate, eIF4A can undergo conformational cycling and alternate between the active-closed and half-open state to enable helicase and ATPase activity [53].

**Figure 3 microorganisms-09-00540-f003:**
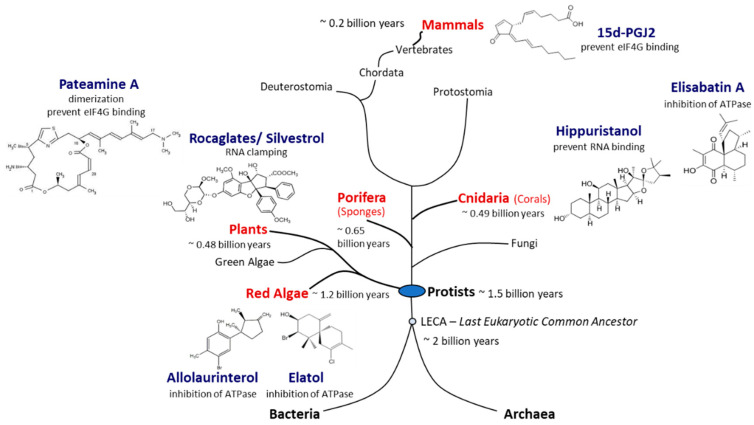
Natural producers of molecules with eIF4A inhibitory activity. Compounds with eIF4A inhibitory activity can be isolated from red algae, plants, sponges, corals, and mammals. Red algae are the phylogenetically most ancient phylum of multicellular eukaryotes for which putative eIF4A inhibitors could be identified. Porifera are the most primordial multicellular animals, and cnidaria are the first animals that developed tissues. Plants and mammals are terrestrial producers of eIF4A inhibitors. Potent eIF4A inhibition has been demonstrated for rocaglates, pateamine A, and hippuristanol. The chemical structures of the inhibitors are diverse and some compounds appear to be straightforward derivatives of other bio-molecules such as sterols in case of hippuristanol or prostaglandins in case of 15d-PGJ2. Although eIF4A inhibitors have not yet been identified in fungi, blocking of eIF4A appears to be a widespread mechanism in eukaryotes.

**Figure 4 microorganisms-09-00540-f004:**
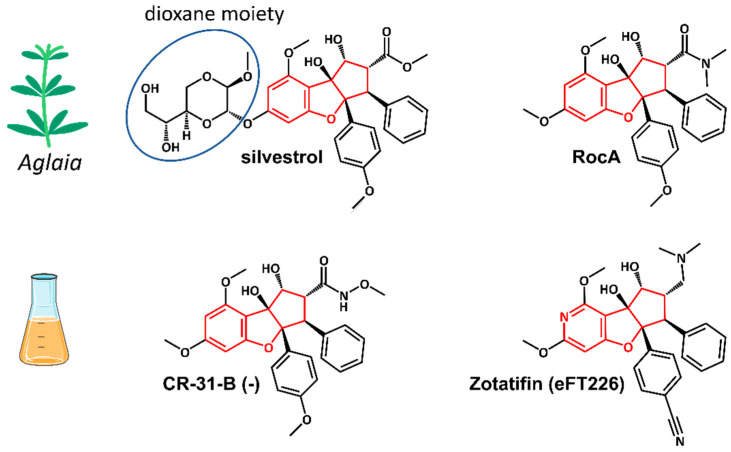
Structures of the natural rocaglates silvestrol and rocaglamide A (RocA) and the synthetic rocaglates CR-31-B (-) and Zotatifin (eFT226). The typical cyclopenta[*b*]benzofurane skeleton is indicated in red. The blue ring shows the dioxane moiety that is only found in silvestrol and its diastereomer episilvestrol.

**Figure 5 microorganisms-09-00540-f005:**
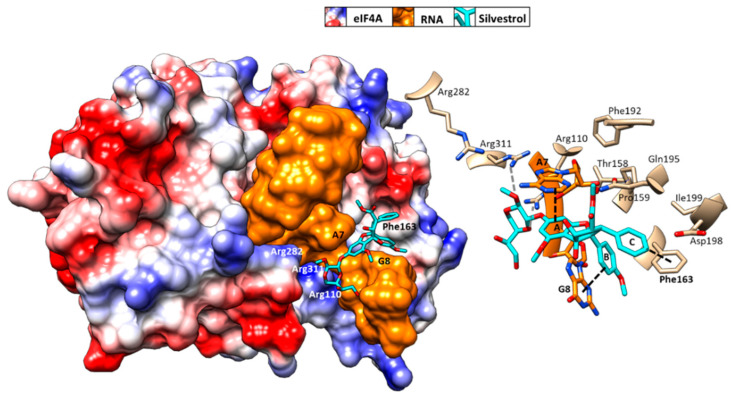
Predicted binding mode of silvestrol using structure-based computational modeling of silvestrol onto the eIF4A–RNA complex (PDB: 5ZC9). The dioxane moiety of silvestrol has the potential to form additional contacts with arginine residues on the surface of eIF4A and may thus bridge over the RNA substrate to tightly clamp the RNA onto eIF4A [83]. UCSF (University of California, San Francisco) Chimera was used for graphical illustration and electrostatic surface coloring of eIF4A (blue: positive charged, red: negative charged).

**Figure 6 microorganisms-09-00540-f006:**
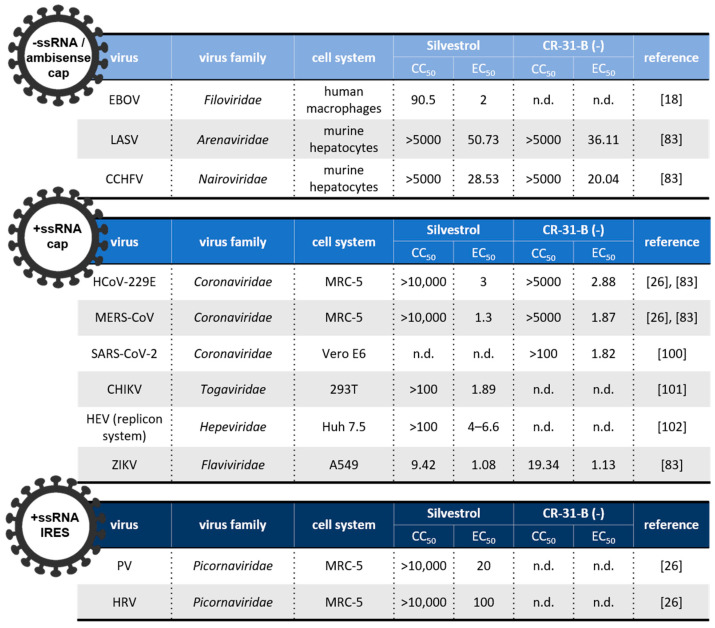
Half maximal cytotoxicity (CC_50_) and effective inhibitory (EC_50_) concentrations (in nM) for cell treatment with silvestrol or CR-31-B (-); CC_50_ values were measured for noninfected (mock-infected) cells and EC_50_ values for cells infected with the indicated viruses [18,26,83,100,101,102]. Ambisense: arenaviruses have negative- and positive-stranded regions in their genomes; cap: 5′ cap-dependent translation initiation; IRES: eIF4A-independent translation initiation via a type III IRES; n.d.: not determined.

**Table 1 microorganisms-09-00540-t001:** Major epidemics and pandemics of zoonotic origin since 2000 (Source: WHO). Abbreviations: COVID-19—coronavirus disease 2019; SARS-CoV-2—severe acute respiratory syndrome coronavirus 2; ZIKV—Zika virus; CHIKV—Chikungunya virus; EBOV—Ebola virus; IAV/H1N1—influenza A virus subtype H1N1.

Years	Disease	Causative Agent	Epidemic/Pandemic	Taxonomic Family
2019–present	COVID-19	SARS-CoV-2	Pandemic	*Coronaviridae*
2015–present	Zika virus disease	ZIKV	Epidemic	*Flaviviridae*
2015/2016	Chikungunya fever	CHIKV	Epidemic	*Togaviridae*
2014–2016	Ebola hemorrhagic fever	EBOV	Epidemic	*Filoviridae*
2012–present	Middle East respiratory syndrome	MERS-CoV	Epidemic	*Coronaviridae*
2009/2010	Influenza	IAV/H1N1	Pandemic	*Orthomyxoviridae*
2002/2003	Severe acute respiratory syndrome	SARS-CoV	Epidemic	*Coronaviridae*

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
