# Peer review of "Targeting the DEAD-Box RNA Helicase eIF4A with Rocaglates—A Pan-Antiviral Strategy for Minimizing the Impact of Future RNA Virus Pandemics"

_microorganisms, 2021, doi:10.3390/microorganisms9030540_

Round 1

Reviewer 1 Report

The manuscript entitled "Targeting the DEAD-box RNA helicase eIF4A with rocaglates…" by Taroncher-Oldenburg et al. reviews recent progress in research of rocaglates, specific eIF4A inhibitors affecting eukaryotic translation initiation, with a special focus on their antiviral properties. Rocaglates are known to clamp eIF4A on mRNA during ribosomal loading and scanning, leading to translation repression (with some selectivity towards mRNA species). The topic is important and timely, as some rocaglates have entered phases I or II of clinical trials as antiviral and/or anticancer drugs. It is also important to note that the rocaglates are very interesting in terms of their molecular mechanism of action.

The review is very well written. Many aspects of rocaglate structural and functional features are carefully discussed. I enthusiastically recommend the manuscript to be published in Microorganisms. To my opinion, it will be equally interesting for scientists studying protein synthesis and for specialists in clinical applications of small molecule drugs.

However, some issues must be improved before the acceptance.

Major point:

Composition of the review looks a little bit strange, as some sections are mixed in a non-optimal way. Specifically, Introduction already contains a substantial portion of information about rocaglates (including chemical structures, Figure 1) and eIF4A-rocaglate interactions (Figure 2). After that the authors move to a description of translation initiation pathway, then to eIF4A once again, and then to eIF4A inhibitors (including the second short description of rocaglates). The next section reviews eIF4A targeting as an antiviral strategy, with subsections devoted to various virus groups (great stuff!), but then the authors go back to mechanism of rocaglate action, followed by a section about clinical applications. To me, it is clear that rocaglate description (containing Figures 1 and 2, as well as information about their mechanism of action, s.6) should be a single section, placed after the section with general description of eIF4A inhibitors (s.3). Thus, section 6 will follow section 4, which will be a much better composition.

Minor points:

  1. Chemical structures n Figure 1 look almost indistinguishable at this resolution. Is it really necessary to show such closely related compounds? I would suggest to show RocA structure instead of episilvestrol, and zotatifin instead of CR-31-B (+).
  2. Throughout the Intro section, a couple of sentences about translation machinery as a promising target for antiviral therapy would be useful (see https://pubmed.ncbi.nlm.nih.gov/33280581/ or some other review for the references of the use of translation inhibitors of various classes to combat viral infections).
  3. Line 149: First, I would remove word “host” here, as a concept of so called “cellular IRESs” is quite questionable, and even if such IRESs do exist, they clearly differ from virus IRESs. Second, I would add “main” to the phrase “two translation mechanisms”, as there are also CITE-directed translation initiation (used by both viral and cellular mRNAs).
  4. Line 178, “three isoforms”: To my opinion, “isoforms” a little bit misleading here, as (1) they are encoded by distinct genes; (2) eIF4AIII is actually quite different protein, in comparison with eIF4AI/II. Please specify (i.e., “three genes encoding distinct eIF4A proteins” or smth like that).
  5. Line 207: Please use superscript in “m7GTP”.
  6. Figure 3. I would suggest to enlarge 4G and 4B a little bit, just to show that they are not small molecules like ATP. Howeve, this is not very important.
  7. Line 238: Even more rocaglates are actually known to date, more than 200 compounds (see, for example, https://pubmed.ncbi.nlm.nih.gov/32101697/).
  8. Line 252: “demonstrated for rocaglates, pateamine A and hippuristanol” – it is better to insert “as well as” before “pateamine A and hippuristanol”, to avoid misunderstanding them as specification of rocaglates.
  9. Line 354: The abbreviation “SFV” is routinely used for Semliki Forest Virus, while CSFV is regularly utilized for the Classical Swine Fever Virus.
  10. Line 434: Please remove the symbol “—“.

Author Response

Point-by-point revision Reviewer #1

Major point:

Composition of the review looks a little bit strange [...] rocaglate description (containing Figures 1 and 2, as well as information about their mechanism of action, s.6) should be a single section, placed after the section with general description of eIF4A inhibitors (s.3). Thus, section 6 will follow section 4, which will be a much better composition.

We have implemented reviewer #1’s suggestion to reorganize the manuscript to improve the flow. The reader now encounters first a description of all known eIF4A inhibitors (section 3), followed by a detailed description of the mechanism of action of rocaglates (section 4) and a section on eIF4A as an antiviral rocaglate target (section 5). We have also adjust the last paragraph of the introduction to reflect the revamped structure of the piece:

“In this review we discuss the essential role of eIF4A in RNA virus translation, the properties of all known eIF4A inhibitors, recent advances in our understanding of the rocaglate-based eIF4A inhibition mechanism, the broad spectrum of rocaglate-mediated eIF4A antiviral activities, and we lay out a roadmap for advancing rocaglates through preclinical and clinical development.”

Minor points:

  1. Chemical structures in Figure 1 look almost indistinguishable at this resolution. Is it really necessary to show such closely related compounds? I would suggest to show RocA structure instead of episilvestrol, and zotatifin instead of CR-31-B (+).

We have added the RocA and Zotatifin structures and removed the episilvestrol and CR-31-B (+) structures from the figure. Note that as a result of the reorganization of the manuscript (see above), this is now Figure 4.

  1. Throughout the Intro section, a couple of sentences about translation machinery as a promising target for antiviral therapy would be useful (see https://pubmed.ncbi.nlm.nih.gov/33280581/ or some other review for the references of the use of translation inhibitors of various classes to combat viral infections).

We have now added the following information into the Introduction (Line 106-108):

“The ongoing SARS-CoV-2 pandemic has further accelerated the clinical development of such antivirals through repurposing compounds already in development to inhibit host translation factors in the context of cancer [20-23].”

  1. Line 149: First, I would remove word “host” here, as a concept of so called “cellular IRESs” is quite questionable, and even if such IRESs do exist, they clearly differ from virus IRESs. Second, I would add “main” to the phrase “two translation mechanisms”, as there are also CITE-directed translation initiation (used by both viral and cellular mRNAs).

We have changed the text (now Line 142) to

“The two translation mechanisms mainly used by RNA viruses are cap-dependent translation and internal ribosome entry site (IRES)-dependent translation.“

  1. Line 178, “three isoforms”: To my opinion, “isoforms” a little bit misleading here, as (1) they are encoded by distinct genes; (2) eIF4AIII is actually quite different protein, in comparison with eIF4AI/II. Please specify (i.e., “three genes encoding distinct eIF4A proteins” or smth like that).

We have now changed the term “isoforms” to “orthologs”. (Line 179)

  1. Line 207: Please use superscript in “m7GTP”.

This has been corrected (Line 208)

  1. Figure 3. I would suggest to enlarge 4G and 4B a little bit, just to show that they are not small molecules like ATP. However, this is not very important

As suggested, 4G and 4B are now enlarged in Figure 2 (formerly Figure 3).

  1. Line 238: Even more rocaglates are actually known to date, more than 200 compounds (see, for example, https://pubmed.ncbi.nlm.nih.gov/32101697/).

We have corrected this information (see Line 247) according to Chu et al., 2020 [78].

  1. Line 252: “demonstrated for rocaglates, pateamine A and hippuristanol” – it is better to insert “as well as” before “pateamine A and hippuristanol”, to avoid misunderstanding them as specification of rocaglates.

We think that this is not required. To insert “as well as” is not grammatically correct at this position in the text.

  1. Line 354: The abbreviation “SFV” is routinely used for Semliki Forest Virus, while CSFV is regularly utilized for the Classical Swine Fever Virus

We have corrected this mistake (Line 432).

  1. Line 434: Please remove the symbol “—“.

This has been done (Line 479).

Reviewer 2 Report

In this review manuscript, the authors describe the possibility to use rocaglates for the development of antiviral strategies for putative future RNA virus pandemics. The rocaglates are compounds that target specifically the RNA helicase eIF4A that is required for scanning of the 5’UTR for efficient translation initiation. The review summarizes the higher sensitivity of RNA virus translation, which therefore makes the rocaglates interesting candidates for anti-viral treatments. Overall, the manuscript is clear, timely, very well written and largely documented by numerous relevant references. Except several minor remarks detailed below, I warmly recommend this manuscript for publication in ‘microorganisms’.

Specific points:

- A general cartoon depicting the translation initiation process at the beginning of the manuscript would greatly benefit to the reader and especially help non-specialists to place the central role of eIF4A in this multi-step and sophisticated process.

- In figure 4, the impact of the detailed inhibitors on eIF4A should be stated in the figure, like ‘preventing binding’, ‘induce 4A/RNA dimerization’ etc … It would also be interesting to map on eIF4A structure 3D, the binding sites of the distinct inhibitors.

- The authors mention the interesting effect of Zotatifin without presenting its structure.

- Section 5 refers to figure 2, therefore the order of figures should adapted to have figure 2 next to section 5.

Author Response

Point-by-point revision Reviewer #2

Specific points:

  1. A general cartoon depicting the translation initiation process at the beginning of the manuscript would greatly benefit to the reader and especially help non-specialists to place the central role of eIF4A in this multi-step and sophisticated process.

We have added a new figure (Figure 1; page 4) depicting the eukaryotic translation initiation mechanism and the role eIF4A plays as part of the translation initiation complex eIF4F.

  1. In figure 4, the impact of the detailed inhibitors on eIF4A should be stated in the figure, like ‘preventing binding’, ‘induce 4A/RNA dimerization’ etc … It would also be interesting to map on eIF4A structure 3D, the binding sites of the distinct inhibitors.

We have added the requested information about the mode-of-action of the known eIF4A inhibitors to Figure 3 (formerly Figure 4). The binding site of rocaglates on eIF4A is shown in Figure 5. Since the focus of our review is on rocaglates as eIF4A inhibitors we think that a detailed scheme showing the potential binding sites of the other eIF4A inhibitors (as far as these binding sites have even been described) is beyond the scope of our article.

  1. The authors mention the interesting effect of Zotatifin without presenting its structure.

The structure of Zotatifin has now been included in Figure 4 (formerly Figure 1).

  1. Section 5 refers to figure 2, therefore the order of figures should adapted to have figure 2 next to section 5.

We have re-structured the manuscript. Figure 2 is now Figure 5 and section 5 (Mechanism of action of rocaglates) is now section 4. The new Figure 5 now correctly refers to the new section 4.

Reviewer 3 Report

In this review, authors provide a detailed and comprehensive analysis of the potential of rocaglates as broad-spectrum antiviral drugs targeting DEAD-box RNA helicase. The paper is well-written, it is clear and in my opinion interesting for medicinal chemists and virologists working in the field.

There are some aspects that could be improved:

1.- In section 3, authors mention a number of eIF4A inhibitors (nicely presented in Figure 4) but it is not clear to me whether any or all of those compounds have an antiviral effect, particularly considering their different molecular scaffolds.  Some discussion should be added to clarify this, even with the possibility of including some graph or Table comparing their inhibitory efficiencies if reliable data are available. 

2.- Considering that zotatifin is an eIF4A inhibitor in clinical trials and a member of the rocaglate family, I think authors should include a Figure with its chemical structure for readers to compare with reference compounds shown in Figure 1.

3.- At the end of section 2, authors mention DDX3 as a related RNA helicase target. Interestingly, recent reports describe new DDX3 inhibitors as broad-spectrum antiviral compounds (see Brai et al. Proceedings of the National academy of sciences USA, 2016, 113, 5388; Brai et al. Journal of medicinal chemistry, 2019, 62, 2333). They should be cited and probably described in the context of the review.

Other comments:

1.- Table 1: It should read “Ebola hemorrhagic fever”. Abbreviations for ZIKV, CHIKV and EBV appear first here and should be defined.

2.- Species names, as well as viral families (Coronaviridae, Togaviridae, etc..), genera, etc… should be written in italics throughout the whole manuscript. Please check out… Examples are: Isis hippuris (line 220), Mycale hentscheli (line 228), etc…

3.- I think that Figure 4 looks better if the rounded box frame is removed.

Author Response

Point-by-point revision Reviewer #3

There are some aspects that could be improved:

  1. In section 3, authors mention a number of eIF4A inhibitors (nicely presented in Figure 4) but it is not clear to me whether any or all of those compounds have an antiviral effect particularly considering their different molecular scaffolds.  Some discussion should be added to clarify this, even with the possibility of including some graph or Table comparing their inhibitory efficiencies if reliable data are available. 

We have added information about the antiviral activities of hippuristanol (Line 231-234) and PatA (Line 241-242) in section 3.

“The antiviral activity of hippuristanol has been documented against several viruses like the encephalomyocarditis virus (EMCV) and the norovirus, two positive-stranded RNA viruses, and human T- cell leukemia virus, type 1 (HTLV-1), a retrovirus [69,72,73].”

“PatA has been shown to have antiviral activity against influenza A virus, a negative-stranded RNA virus [76].”

The antiviral activity of rocaglates is extensively discussed in section 5 “eIF4A as an antiviral rocaglate target” (formerly section 4). For the other compounds to our knowledge no comparative studies exist where direct antiviral activities were analyzed. A comparative graph or a table showing the respective EC50 values for eIF4A inhibitors regarding antiviral activities is therefore not possible.

  1. Considering that zotatifin is an eIF4A inhibitor in clinical trials and a member of the rocaglate family, I think authors should include a Figure with its chemical structure for readers to compare with reference compounds shown in Figure 1.

We have now included the structure of Zotatifin in Figure 4 (formerly Figure 1).

  1. At the end of section 2, authors mention DDX3 as a related RNA helicase target. Interestingly, recent reports describe new DDX3 inhibitors as broad-spectrum antiviral compounds (see Brai et al. Proceedings of the National academy of sciences USA, 2016, 113, 5388; Brai et al. Journal of medicinal chemistry, 2019, 62, 2333). They should be cited and probably described in the context of the review.

More information about DDX3 inhibitors with broad-spectrum antiviral activities have been included in the text body (see last paragraph section 2, Line 210-213). The requested papers from Brai et al., [58-60] are now cited.

“DDX3 also plays a role equivalent to that of eIF4A in class I and II IRES-dependent translation [45], and its essential role in viral RNA translation has been documented for a number of viruses including the RNA viruses Japanese encephalitis virus, Dengue virus (DENV), and West Nile virus [55–58]. Several studies have further shown the broad-spectrum antiviral potential of targeting DDX3 [59–61].”

Other comments:

  1. Table 1: It should read “Ebola hemorrhagic fever”. Abbreviations for ZIKV, CHIKV and EBV appear first here and should be defined.

The mistakes in Table 1 are corrected (see page 2). Abbreviations for the mentioned viruses have been defined in the legend of Table 1.

Table 1. Major epidemics and pandemics of zoonotic origin since 2000 (Source: WHO). Abbreviations: COVID-19 – coronavirus disease 2019; SARS-CoV-2 - severe acute respiratory syndrome coronavirus 2; ZIKV – Zika virus; CHIKV – Chikungunya virus; EBOV – Ebola virus; IAV/H1N1 – influenza A virus subtype H1N1.

  1. Species names, as well as viral families (Coronaviridae, Togaviridae, etc..), genera, etc… should be written in italics throughout the whole manuscript. Please check out… Examples are: Isis hippuris (line 220), Mycale hentscheli (line 228), etc…

We have checked the text and now all species names and viral families are written in italics throughout the manuscript.

  1. I think that Figure 4 looks better if the rounded box frame is removed.

The rounded box frame has been removed from Figure 3 (formerly Figure 4, see page 7).

Round 2

Reviewer 1 Report

The manuscript has been largely improved and now can be accepted and published in its present form, except a minor correction: please replace "orthologs" with "paralogs" (regarding three eIF4A homologs, line 179). There is no need for another round of sending it to referee(s).

Best wishes,

S.D.

Author Response

Dear Reviewer,

In the revised version of the manuscript (minor revision) the term "orthologs" has been replaced by "paralogs" (Line 179).

The manuscript is now in a clean version. The formerly marked changes in red are now notated in black.